# Mucinous Ovarian Carcinoma: Integrating Molecular Stratification into Surgical and Therapeutic Management

**DOI:** 10.3390/biomedicines13051198

**Published:** 2025-05-14

**Authors:** Mauro Francesco Pio Maiorano, Brigida Anna Maiorano, Gennaro Cormio, Vera Loizzi

**Affiliations:** 1Unit of Obstetrics and Gynecology, Department of Interdisciplinary Medicine (DIM), University of Bari “Aldo Moro”, Polyclinic of Bari, Piazza Giulio Cesare 11, 70124 Bari, Italy; mauro.maiorano95@outlook.it (M.F.P.M.); gennaro.cormio@uniba.it (G.C.); 2Unit of Oncologic Gynecology, IRCCS “Giovanni Paolo II” Oncologic Institute, Viale Orazio Flacco 65, 70124 Bari, Italy; vera.loizzi@uniba.it; 3Department of Medical Oncology, IRCCS San Raffaele Hospital, Via Olgettina 60, 20132 Milan, Italy; 4Translational Biomedicine and Neuroscience Department (DiBraiN), University of Bari “Aldo Moro”, Piazza Giulio Cesare 11, 70124 Bari, Italy

**Keywords:** mucinous ovarian carcinoma, mucinous ovarian cancer treatment, mucinous ovarian tumor, mucinous ovarian cancer adjuvant therapy, ovarian cancer fertility sparing surgery, mucinous ovarian cancer KRAS, mucinous ovarian cancer HER2, mucinous ovarian cancer staging, mucinous ovarian carcinoma histology, mucinous ovarian carcinoma chemotherapy

## Abstract

**Background/Objectives**: Mucinous ovarian carcinoma (MOC) is a rare and biologically distinct subtype of epithelial ovarian cancer, typically presenting at an early stage in younger women. Unlike high-grade serous carcinoma, MOC is characterized by unique molecular features—including frequent KRAS mutations and HER2 amplifications—and exhibits limited sensitivity to platinum-based chemotherapy. These differences highlight the need for individualized treatment strategies guided by molecular and histological profiling. This review aims to integrate current evidence on the clinical management of MOC with emerging insights into its molecular biology, with a focus on how these factors influence surgical decision-making, fertility preservation, and adjuvant therapy selection. **Methods**: We performed a comprehensive narrative review of the literature, synthesizing findings from retrospective cohorts, molecular studies, and clinical guidelines relevant to the surgical, reproductive, and therapeutic management of MOC. **Results**: Histologic subtype—expansile versus infiltrative—plays a critical role in guiding lymphadenectomy as lymph node metastases are rare (<1%) in expansile tumors but occur in up to 23% of infiltrative cases. Complete surgical staging remains essential for accurate prognostication, yet tailored approaches may reduce overtreatment in low-risk patients. Fertility-sparing surgery (FSS) appears safe in FIGO stage IA expansile MOC, with favorable reproductive outcomes, while higher-stage or infiltrative cases warrant caution. Given MOC’s chemoresistance, the role of adjuvant therapy in early-stage disease remains debated. Targeted strategies, including MEK inhibitors and HER2-directed therapies, are under investigation and may benefit selected molecular subgroups. **Conclusions**: MOC requires a nuanced, biomarker-informed approach. This review advocates for personalized, evidence-based management supported by multidisciplinary evaluation while underscoring the urgent need for prospective studies and biomarker-driven clinical trials.

## 1. Introduction

Mucinous ovarian carcinoma (MOC) is a rare histological subtype of epithelial ovarian cancer (EOC), accounting for approximately 3–5% of all cases [1]. It is also the most frequent histological subtype in women under 40 [2]. Primary MOCs are thought to arise from either gastrointestinal-type epithelium or endocervical-like epithelium, with the former being more common [3]. These origins are reflected in their distinct molecular profiles and patterns of metastatic spread. The histopathological classification of MOC has evolved significantly, with a critical distinction between borderline ovarian tumors (BOT) and invasive MOC [4]. While BOTs exhibit epithelial proliferation without stromal invasion, invasive MOCs demonstrate unequivocal stromal invasion, often reflecting more aggressive clinical behavior [4]. The intestinal-type MOC is more common and includes two distinct growth patterns: expansile and infiltrative. Expansile MOC shows a confluent glandular growth pattern with minimal to no stromal invasion, whereas infiltrative MOC is characterized by destructive stromal invasion with clusters of malignant cells [3]. Distinct from high-grade serous ovarian carcinoma (HGSOC), MOC is characterized by frequent Kirsten rat sarcoma viral oncogene homolog (KRAS) gene mutations and human epidermal growth factor receptor 2 (HER2) amplification—and lacks the widespread genomic instability and TP53 mutations typical of HGSOC [5,6,7,8]. These molecular features correlate with markedly different clinical behavior, including a low propensity for peritoneal dissemination, a high rate of early-stage diagnosis (70–80% at FIGO stage I–II), and limited responsiveness to platinum-based chemotherapy [9,10,11]. Indeed, current molecular evidence has shown that MOC with an infiltrative growth pattern tends to be CK5/6, CD24, and EGFR positive, suggesting that these markers may be linked to its aggressive features [12]. In contrast, expansile invasion showed a higher prevalence of HER2 overexpression/amplification and less frequent HER2 mutation compared to infiltrative MOC [13]. Additionally, PAX8 expression was more commonly associated with expansile invasion [13]. Due to its rarity, most of the available data on MOC are derived from studies primarily focused on EOC, where mucinous cases constitute only a small subset (typically <10% of study populations) [14]. As a result, treatment strategies and surgical management protocols remain a topic of ongoing debate. Historically, surgical staging has been a cornerstone of treatment for early-stage MOC, with peritoneal and lymph node assessment playing a crucial role in determining the extent of the disease [15]. However, given the low prevalence of lymphatic spread in early-stage MOC, the necessity of routine lymphadenectomy has been questioned [16]. Studies suggest that lymph node metastases are primarily observed in the infiltrative subtype, while the expansile subtype rarely exhibits nodal involvement [16,17]. This distinction underscores the importance of histologic evaluation in guiding surgical decisions as infiltrative MOC may require more extensive staging. In contrast, expansile MOC might be safely managed with a more conservative approach. The distinction between expansile and infiltrative growth patterns is pivotal for histopathologic classification and guiding surgical strategy, risk assessment, and fertility preservation. The table below summarizes the key clinical, prognostic, and molecular features differentiating the two subtypes (Table 1).

Another critical aspect of MOC management is the role of fertility-sparing surgery (FSS) in young patients. Given the favorable prognosis of stage I MOC, with 5-year overall survival (OS) rates exceeding 90%, FSS has been explored as a viable option for preserving reproductive potential [14,17,18,19]. While evidence suggests that oncologic outcomes following FSS are comparable to those of radical surgery in well-selected cases, concerns persist regarding recurrence rates, particularly in patients with incomplete surgical staging where disease-free survival (DFS) is significantly lower [20,21,22]. Despite advancements in understanding MOC, significant gaps remain in defining the optimal surgical approach, the necessity of adjuvant therapy, and the oncologic safety of FSS. This review aims to synthesize current evidence on the surgical and therapeutic management of MOC, integrating molecular and histopathologic data to inform lymphadenectomy decisions, fertility-sparing approaches, and the evolving role of targeted therapy in this biologically distinct malignancy.

## 2. Surgical and Therapeutic Management of Mucinous Ovarian Carcinoma: A Comprehensive Review

### 2.1. Surgical Staging and the Role of Lymphadenectomy

Surgical staging is a critical component of the management of MOC, particularly for early-stage disease. Unlike HGSOC, MOC is less likely to present with peritoneal dissemination, with lymphatic spread reported in only 0.8% to 17% of cases [17]. However, studies suggest that the risk of lymph node metastases differs significantly between histologic subtypes, with it being higher in infiltrative MOC [16,17]. Recent evidence from a systematic review and meta-analysis demonstrated that expansile MOC is predominantly associated with early-stage presentation (89.8% at FIGO stage I) and a lower recurrence rate (6.9%), while infiltrative MOC frequently presents at more advanced stages (only 56.2% at FIGO stage I) and exhibits a substantially higher recurrence rate (24.5%). Moreover, the death rate is significantly lower for expansile MOC (10.5%) compared to infiltrative MOC (31.1%) [18]. Together these findings suggest that systematic lymphadenectomy may not be necessary for expansile MOC, where the likelihood of nodal disease is minimal [16,17]. In a meta-analysis by Hoogendam et al., the pooled prevalence of lymph node metastases among apparent stage I–II expansile MOC patients was only 0.8% (95% CI < 0.1–2.9%), with no difference between nodal sampling and full dissection [23]. This strongly supports a conservative approach in this subtype. Conversely, infiltrative MOC, particularly high-grade tumors, may warrant lymphadenectomy for accurate staging and prognostication [17]. A multicenter study involving 94 women with MOC reported no cases of nodal metastases in patients with the expansile subtype [24]. Conversely, the infiltrative subtype demonstrates a higher propensity for lymphatic spread; in the same study, approximately 23% of patients with infiltrative MOC had metastatic involvement in the pelvic and/or para-aortic lymph nodes [24]. These findings are consistent with other research. A study analyzing 107 patients with primary mucinous ovarian carcinomas found no cases of lymph node metastasis among 51 patients who underwent lymphadenectomy for tumors grossly confined to the ovary [25]. Current recommendations suggest that lymphadenectomy should be performed selectively based on histologic subtype and tumor grade rather than as a universal standard [26]. The 2024 ESGO-ESMO-ESP guidelines emphasize a risk-adapted approach, mirroring strategies used for other non-serous ovarian cancers [26]. Given the challenges in accurately determining the invasive pattern intraoperatively, an individualized approach to lymphadenectomy is advisable. Preoperative imaging, intraoperative assessment, and—where available—molecular profiling (KRAS or HER2 status) may aid in identifying patients who could safely forgo extensive nodal dissection and should guide the decision, balancing the potential benefits of lymph node dissection against the risks associated with the procedure.

### 2.2. The Importance of Complete Surgical Staging

While MOC is more frequently diagnosed at early stages (International Federation of Gynecology and Obstetrics (FIGO)-staging system: I–II in 70–80% of cases), the potential for occult metastases highlights the need for adequate surgical staging. In a study evaluating staging completeness, 9.2% of patients with apparent FIGO stage I disease were upstaged to FIGO stage II-IVB, particularly in cases with bilateral ovarian involvement [27]. This underscores the importance of meticulous intraoperative assessment, including peritoneal cytology and omental biopsy, to ensure that patients are not under-staged. However, given the low rate of peritoneal involvement compared to HGSOC, some authors have suggested that less aggressive staging strategies may be appropriate for low-risk MOC cases [10,11,15]. The feasibility of selective staging approaches remains an area of ongoing investigation, and future research should focus on identifying clinical and molecular markers that predict the risk of occult metastases. A critical aspect of surgical management is the prevention of intraoperative tumor rupture. Historically, such ruptures have been associated with the dissemination of malignant cells into the peritoneal cavity, potentially increasing the risk of metastasis and recurrence, thereby adversely affecting prognosis. However, recent studies have provided nuanced insights into this issue. For instance, a retrospective analysis involving 170 women with MOC found that intraoperative rupture occurred in 35% of cases [28]. Notably, this study concluded that intraoperative rupture did not independently confer a worse survival outcome, suggesting that the prognostic impact of rupture may be less significant than previously thought [28]. Conversely, a large-scale study analyzing data from 15,163 women with stage IA-IC1 EOC indicated that the prognostic significance of intraoperative capsule rupture varies across histological subtypes [29]. Specifically, clear cell histology exhibited the most pronounced adverse effect on survival following rupture, whereas the impact on mucinous tumors was comparatively less severe [29]. Given these findings, while the prevention of intraoperative rupture remains a surgical priority its occurrence should prompt a comprehensive evaluation of individual patient factors, tumor histology, and other prognostic indicators to guide postoperative management decisions effectively.

### 2.3. Fertility-Sparing Surgery in MOC

Given that MOC often affects younger women, fertility preservation is a key consideration in its surgical management. FSS has been explored in well-selected stage I patients, with high reproductive success rates [17,22]. Studies report that 91.3% of patients attempting pregnancy conceived successfully, with a live birth rate of 88.9% [22]. From an oncologic perspective, DFS following FSS appears comparable to radical surgery in low-risk patients, with 5-year DFS rates of 82.5% (FSS) vs. 94.5% (radical surgery), *p* = 0.207 [22]. However, recurrence risk persists, particularly in FIGO IC disease where survival outcomes may be compromised [22,30]. Additionally, incompletely staged patients had significantly lower DFS (66.7% vs. 85.0%), reinforcing the need for comprehensive assessment before offering conservative surgery [22,27]. While available data support the safety of FSS in expansile MOC and FIGO IA disease, its role in infiltrative subtypes remains uncertain. Some case reports suggest that even selected infiltrative MOC patients may achieve long-term recurrence-free survival with FSS, but larger studies are needed to define appropriate patient selection criteria [30]. Studies focusing on specific molecular alterations, such as tumor protein 53 (TP53), KRAS mutation, or HER2 amplification could further guide the decision regarding FSS feasibility and safety.

To support individualized surgical decision-making, we developed a biomarker-informed algorithm that integrates current evidence on histologic subtype, nodal metastasis risk, fertility preservation, and the role of adjuvant treatment in MOC, incorporating both clinical and molecular considerations (Figure 1).

### 2.4. Oncologic Outcomes and Prognostic Factors

The prognosis of MOC differs significantly from HGSOC, with early-stage disease exhibiting 5-year OS rates exceeding 90% [5,10]. However, survival outcomes worsen in advanced-stage MOC, largely due to its poor response to chemotherapy and the impact of residual disease after surgery [5]. The most important prognostic factors include tumor histology (expansile vs. infiltrative subtypes), stage at diagnosis (FIGO I vs. FIGO II-IVB), extent of cytoreductive surgery (residual tumor-R—0 vs. R1/R2 resections), and lymph node involvement (more common in infiltrative MOC) [5,10]. Given its distinct behavior, MOC requires a surgical-first approach, prioritizing complete resection (R0) whenever possible [5,31]. Unlike HGSOC, where neoadjuvant chemotherapy (NACT) followed by interval debulking is standard for advanced disease, MOC is often chemoresistant, making primary debulking surgery the preferred strategy whenever feasible [31].

### 2.5. The Role of Adjuvant Therapy in Early-Stage MOC

The benefit of adjuvant chemotherapy in early-stage MOC remains controversial. In stage I disease, retrospective data suggest that chemotherapy does not significantly improve survival, with a 10-year OS of 79% in non-chemotherapy patients vs. 81% in chemotherapy-treated patients (*p* = 0.46) [32]. For instance, a study utilizing the U.S. National Cancer Database evaluated 4811 patients with stage I MOC and found that 30.9% received adjuvant chemotherapy [33]. The 5-year OS rate was 86.8% for those who received chemotherapy compared to 89.7% for those who did not, with no statistically significant difference between the groups (*p* = 0.17) [33]. However, in high-risk subgroups (FIGO IC, grade 3 tumors, or infiltrative histology), chemotherapy has been associated with a 23% improvement in long-term survival (51% vs. 74%, HR = 1.58, *p* = 0.03) [33]. Given MOC’s intrinsic resistance to platinum-based chemotherapy, alternative strategies are being explored. HER2-targeted therapy (trastuzumab) and mitogen-activated protein kinase-MAPKK (MEK) inhibitors (for KRAS-mutated tumors) represent promising approaches for selected patients, though further clinical validation is needed [34,35,36,37,38,39,40].

To support the recommendations discussed, we summarized key findings from recent retrospective studies on mucinous ovarian carcinoma, including those addressing lymphadenectomy, fertility-sparing surgery, adjuvant chemotherapy, and genomic profiling (Table 2)

### 2.6. Molecular Considerations and Emerging Targeted Therapies in Mucinous Ovarian Carcinoma

MOC is characterized by distinct molecular alterations that have prompted the exploration of targeted therapeutic strategies. Among these, mutations in the KRAS gene are notably prevalent, occurring in approximately 65.8% of cases [35]. These mutations lead to the continuous activation of the mitogen-activated protein kinase (MAPK) signaling pathway, promoting tumor cell proliferation and survival [41]. Given the high frequency of KRAS mutations in MOC, targeted therapies have been a focus of recent research [35]. MEK inhibitors, which act downstream of KRAS in the MAPK pathway, have shown potential in preclinical studies [42]. For instance, trametinib, a MEK inhibitor, has demonstrated efficacy in reducing tumor growth in KRAS-mutated ovarian cancer models [43]. Another significant molecular alteration in MOC is the amplification of the ERBB2 gene, observed in approximately 26.7% of cases [44]. This amplification leads to the overexpression of the HER2 protein, providing a rationale for the use of HER2-targeted therapies [45]. Trastuzumab, a monoclonal antibody against HER2, has been investigated in this context. A study reported that patients with HER2-overexpressing MOC treated with trastuzumab-containing regimens experienced improved progression-free survival compared to those receiving standard chemotherapy [34]. Despite these promising avenues, the clinical application of targeted therapies in MOC faces challenges. The rarity of MOC limits the feasibility of large-scale clinical trials, and the heterogeneity of molecular alterations necessitates personalized treatment approaches. Furthermore, while agents like sotorasib have been approved for KRAS G12C-mutated non-small-cell lung cancer, their efficacy in KRAS-mutated MOC remains under investigation [46].

In conclusion, the unique molecular landscape of MOC underscores the potential of targeted therapies. Ongoing research into agents targeting KRAS mutations and HER2 amplifications holds promise for improving outcomes in patients with this challenging malignancy. However, further clinical studies are essential to validate these approaches and to develop effective, individualized treatment strategies for MOC.

## 3. Discussion

The evolving management of MOC underscores the increasing importance of histologic precision and molecular stratification in gynecologic oncology. While surgical resection remains the foundation of curative treatment, particularly in early-stage disease, mounting evidence suggests that decision-making should integrate not only tumor stage and histotype but also underlying molecular alterations. The experience gained from the molecular reclassification of endometrial carcinoma by The Cancer Genome Atlas (TCGA)—which redefined prognostic categories and therapeutic targets—might serve as a valuable model for MOC, a tumor long grouped under the broad umbrella of EOC despite its distinct pathogenesis and clinical behavior [47]. Current data support a conservative surgical approach in expansile MOC, which is characterized by a minimal risk of lymphatic dissemination. In contrast, the infiltrative subtype often necessitates complete staging and the consideration of adjuvant therapy. The oncologic safety of FSS has been validated in well-selected expansile MOC patients, especially FIGO stage IA, with high reproductive success and comparable disease-free survival. However, recurrence remains a concern in incompletely staged or infiltrative cases, and in these histologic and molecular markers should guide caution. MOC’s poor response to platinum-based chemotherapy, even in advanced disease, further supports the need for a biomarker-driven shift in systemic treatment paradigms. In this context, molecular profiling reveals therapeutic vulnerabilities and additional alterations (i.e., RNF43, PIK3CA, ARID1A, and BRAF [including V600E]) have been reported, emphasizing the complexity and heterogeneity of MOC and pointing to future avenues for individualized therapy [48]. Despite these promising insights, the clinical translation of molecular findings into routine practice remains limited by the rarity of MOC, underrepresentation in prospective trials, and a lack of standardized diagnostic and therapeutic algorithms [47]. Molecular-informed subclassification could enhance prognostic modeling, refine eligibility for conservative approaches, and guide inclusion in targeted therapy trials. Furthermore, integrating next-generation sequencing and biomarker panels into early diagnostic workup may improve subtype recognition and therapeutic planning. In summary, MOC demands a departure from conventional ovarian cancer management. Future research should prioritize prospective, biomarker-annotated trials to validate molecular classifiers and define targeted therapeutic efficacy. Until such data mature, an individualized, multidisciplinary strategy that integrates histopathology, molecular biology, fertility desires, and surgical expertise remains essential to optimize outcomes in this rare but biologically distinct malignancy.

## 4. Conclusions

MOC represents a rare and biologically distinct subtype of ovarian cancer that requires a departure from traditional EOC treatment paradigms. Histologic subtype and molecular profiling are essential to guide surgical staging, fertility-sparing decisions, and systemic therapy. Advancing the management of MOC will depend on biomarker-driven research, collaborative clinical trials, and the integration of precision oncology into routine care.

## Figures and Tables

**Figure 1 biomedicines-13-01198-f001:**
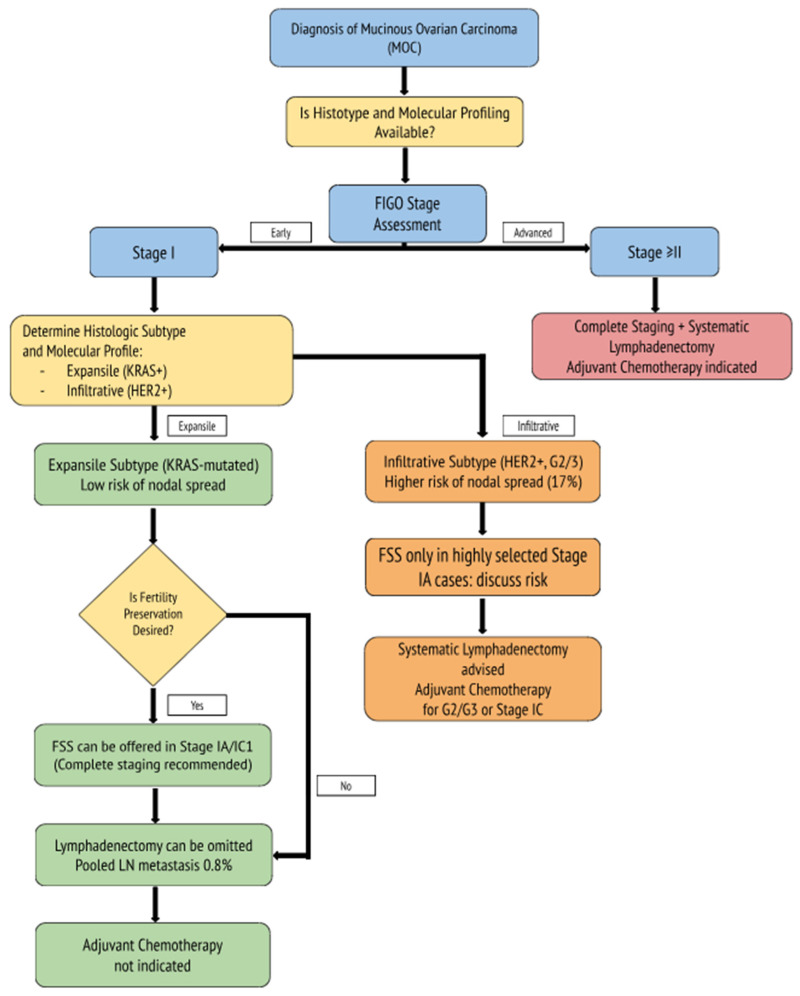
Molecularly informed clinical algorithm for MOC. Management is tailored based on histologic subtype (expansile vs. infiltrative) and molecular markers (e.g., KRAS and HER2 status). Expansile tumors, often KRAS-mutated and low-risk for nodal spread, may be managed conservatively with FSS and the omission of lymphadenectomy. Infiltrative tumors, frequently HER2-amplified and higher grade, warrant more extensive staging and consideration of adjuvant chemotherapy, even in early-stage disease.

**Table 1 biomedicines-13-01198-t001:** Comparative overview of expansile and infiltrative MOC. Expansile tumors, which are less commonly encountered, carry a lower risk of lymph node involvement and generally have a more favorable prognosis, often allowing for fertility-sparing approaches. In contrast, infiltrative MOC is associated with a higher rate of nodal metastases and a more guarded prognosis, requiring more extensive surgical staging. Differences in molecular alterations are also shown, and their integration may guide future clinical decisions [1,2,3,4,5,6,7,8,9,10,11,12,13,14,15,16,17].

Feature	Expansile MOC	Infiltrative MOC
Frequency	43%	57%
Lymph Node Metastasis Rate	0–1%	~17%
Lymphadenectomy	Often omitted	Recommended
Prognosis	Generally favorable	Less favorable
FSS Suitability	High (IA)	Controversial/Selected IA
Chemotherapy Sensitivity	Low	Low
KRAS Mutation	~70–85%	~40–60%
HER2 Amplification	~20–30%	>30%
TP53 Mutation	Rare	May occur
MUC1/MUC16 Overexpression	Common	Common

**Table 2 biomedicines-13-01198-t002:** Summary of key retrospective and molecular studies on MOC, highlighting data on the histologic subtype, nodal involvement, fertility outcomes, adjuvant therapy efficacy, and molecular alterations potentially relevant to clinical decision-making.

Study (Year)	Sample Size	Outcome(s) and Molecular Insights	Key Results
Chen et al. (2025) [18]	1185	Prognostic value of growth pattern-based grading in MOC.	Expansile MOC: Death rate 10.5%, Recurrence rate 6.9%, and FIGO I rate 89.8%. Infiltrative MOC: Death rate 31.1%, Recurrence rate 24.5%, and FIGO I rate 56.2%. Infiltrative pattern linked to poorer prognosis. Complete surgical staging recommended for infiltrative MOC.
Richardson et al. (2020) [32]	2041 patients	Impact of adjuvant chemotherapy on survival in stage I MOC.	10-year OS rate: 79% (no CHT) vs. 81% (CHT). CHT improved survival only in high-risk patients (HR = 1.58, *p* = 0.03).
Gouy et al. (2017) [17]	68 patients (29 expansile and 39 infiltrative)	Lymphadenectomy necessity, peritoneal spread, and upstaging rates in early-stage MOC.	Lymphadenectomy in 31 patients (8 expansile, 23 infiltrative). Nodal metastases in four infiltrative cases (17%). Microscopic peritoneal spread in two cases. One patient upstaged from IA to IC3 due to positive cytology.
Huin et al. (2022) [24]	94 patients (35 expansile and 59 infiltrative)	Clinical presentation and prognosis by histologic subtype.	Lymph node metastases: 21% in infiltrative vs. 0% in expansile. 5-year recurrence-free survival (RFS): 90% (expansile) vs. 60% (infiltrative). Adjuvant chemotherapy used in 46% of infiltrative vs. 20% of expansile cases.
Schmeler et al. (2010) [25]	107 patients with primary MOC (51 with lymphadenectomy)	Prevalence of lymph node metastases and staging outcomes in early-stage MOC.	51 patients with tumors confined to the ovary: 0% nodal metastases. No significant difference in 5-year OS (83% vs. 69%) or PFS (80% vs. 63%) between patients with and without lymphadenectomy. Routine lymphadenectomy may be omitted in clinically early-stage MOC.
Yuan et al. (2022) [27]	163 patients	Upstaging rates after complete surgical staging.	9.2% upstaged to FIGO stage II-IVB; risk factors: bilateral ovarian involvement (OR = 9.739, *p* = 0.005) and history of MOC (OR = 4.745, *p* = 0.033).
Lin et al. (2022) [22]	159 patients	Oncologic and reproductive outcomes after FSS.	5-year DFS rate: 82.5% (FSS) vs. 94.5% (RS) (*p* = 0.207). Pregnancy success rate 91.3%. Live birth rate 88.9%.
Frumovitz et al. (2010) [40]	Multiple cohorts	Prognosis and treatment response KRAS ~50%, TP53 ~16%, and BRCA <2%.	Poor CHT response; OS in advanced MOC: ~12–15 months vs. ~36–45 in serous carcinoma.
Cheasley et al. (2019) [8]	255 MOC cases (134 sequenced)	KRAS/TP53 64%, HER2 amp 26%, and CDKN2A loss 76%.	Copy number burden linked to high-grade progression.

BRCA: breast cancer-associated gene; CDKN2A: cyclin-dependent kinase inhibitor 2A; CHT: chemotherapy; DFS: disease-free survival; FIGO: International Federation of Gynecology and Obstetrics; FSS: fertility-sparing surgery; HER2: human epidermal growth factor receptor 2; HR: hazard ratio; KRAS: Kirsten rat sarcoma viral oncogene homolog; LN: lymph node; MOC: mucinous ovarian carcinoma; OR: odds ratio; OS: overall survival; PFS: progression-free survival; RFS: recurrence-free survival; RS: radical surgery; TP53: tumor protein p53.

## Data Availability

The authors confirm that the data supporting the findings of this study are available within the article.

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
