# Peer review of "Mucinous Ovarian Carcinoma: Integrating Molecular Stratification into Surgical and Therapeutic Management"

_biomedicines, 2025, doi:10.3390/biomedicines13051198_

Round 1
Reviewer 1 Report
Comments and Suggestions for Authors
This review article provides a comprehensive discussion on the management and molecular profile of mucinous ovarian carcinoma (MOC). The manuscript is well written and provides a good summary of the established management and current genetic investigations on ovarian MOCs. The following are suggested to further improve this paper:
- In the introduction, please provide a statement on the histopathologic origins of primary ovarian MOCs.
- Table 1 needs references.
Author Response
We would like to thank the reviewer for the precious comments, which greatly contributed to improving our manuscript, which we hope will now meet the standard for publication.
Comment 1: In the introduction, please provide a statement on the histopathologic origins of primary ovarian MOCs.
Response 1: Thank you, we revised the section accordingly and added the statement.
Comment 2: Table 1 needs references.
Response 2: Thank you. We added references both in Table 1's caption and within Table 2, for clarity.
Thank you again for your valuable time.
Dr. Brigida Anna Maiorano,
on behalf of all authors.
Reviewer 2 Report
Comments and Suggestions for Authors
This review aims to characterize the management of mucinous ovarian carcinoma (MOC) based on growth pattern type (expansile vs. infiltrative), with a focus on surgical staging, fertility-sparing surgery (FSS), and adjuvant treatment.
While it is generally well written, it omits a recently published and highly relevant study: Chen M, et al. The prognostic value of growth pattern-based grading for mucinous ovarian carcinoma (MOC): a systematic review and meta-analysis. Front Oncol. 2025 Mar 31;15:1541572. doi:10.3389/fonc.2025.
This meta-analysis provides the highest level of evidence currently available on this topic and offers clearer guidance for the treatment and clinical management of MOC. Its inclusion is essential and should take precedence over several smaller retrospective observational studies currently cited.
It would be valuable to include in the Background section a description of the evolution of the pathological classification of mucinous tumors—particularly the distinction between borderline ovarian tumors (BOT) and mucinous ovarian carcinoma (MOC), the subtypes of primary MOC (intestinal vs. endocervical-type), and within the intestinal type, the expansile vs. infiltrative growth patterns—along with current molecular characteristics.
The Background, Results, and Discussion sections contain several instances of repetition, which should be revised for clarity and conciseness
Author Response
We would like to thank the reviewer for the precious comments, which greatly contributed to improving our manuscript, which we hope will now meet the standard for publication.
Comment 1:
While it is generally well written, it omits a recently published and highly relevant study: Chen M, et al. The prognostic value of growth pattern-based grading for mucinous ovarian carcinoma (MOC): a systematic review and meta-analysis. Front Oncol. 2025 Mar 31;15:1541572. doi:10.3389/fonc.2025. This meta-analysis provides the highest level of evidence currently available on this topic and offers clearer guidance for the treatment and clinical management of MOC. Its inclusion is essential and should take precedence over several smaller retrospective observational studies currently cited.
Response 1: Thank you for pointing this out. This study had not yet been published at the time of data extraction. We added and discussed this valuable study in the review section and Table 2.
Comment 2: It would be valuable to include in the Background section a description of the evolution of the pathological classification of mucinous tumors—particularly the distinction between borderline ovarian tumors (BOT) and mucinous ovarian carcinoma (MOC), the subtypes of primary MOC (intestinal vs. endocervical-type), and within the intestinal type, the expansile vs. infiltrative growth patterns—along with current molecular characteristics.
Response 2: Thank you. We addressed these valuable points in our Introduction section and added references.
Comment 3: The Background, Results, and Discussion sections contain several instances of repetition, which should be revised for clarity and conciseness.
Response 3: We generally revise these sections accordingly.
Thank you for your valuable time.
Dr. Brigida Anna Maiorano,
on behalf of all co-authors